

# Lightweight data management with dtool

Tjelvar S.G. Olsson[*] and Matthew Hartley[*]

Computational Systems Biology, John Innes Centre, Norwich, UK, United Kingdom
[*] These authors contributed equally to this work.

## ABSTRACT

The explosion in volumes and types of data has led to substantial challenges in data management. These challenges are often faced by front-line researchers who are already dealing with rapidly changing technologies and have limited time to devote to data management. There are good high-level guidelines for managing and processing scientific data. However, there is a lack of simple, practical tools to implement these guidelines. This is particularly problematic in a highly distributed research environment where needs differ substantially from group to group and centralised solutions are difficult to implement and storage technologies change rapidly. To meet these challenges we have developed dtool, a command line tool for managing data. The tool packages data and metadata into a unified whole, which we call a dataset. The dataset provides consistency checking and the ability to access metadata for both the whole dataset and individual files. The tool can store these datasets on several different storage systems, including a traditional file system, object store (S3 and Azure) and iRODS. It includes an application programming interface that can be used to incorporate it into existing pipelines and workflows. The tool has provided substantial process, cost, and peace-of-mind benefits to our data management practices and we want to share these benefits. The tool is open source and available freely online at http://dtool.readthedocs.io.

## INTRODUCTION

Science is an empirical discipline and therefore requires careful data management. Advances in our ability to capture and store data have resulted in a "big data explosion". This is particularly true in biology and has resulted in data management becoming one of the big challenges faced by the biological sciences (*Howe et al., 2008*; *Stephens et al., 2015*; *Cook et al., 2018*).

Data management is a broad term and means different things to different people. At a high level, funders and the research community care about data being trusted, shared and reusable (*Vision, 2010*; *Wilkinson et al., 2016*; *Waard, Cousijn & Aalbersberg, 2018*, *Leek, 2018*). At an intermediate level, research institutes and principal investigators need to think about the life cycle of data (*Lynch, 2008*; *Michener, 2015*), and how to get the resources they need for it. At the ground level individual researchers need to think about how to structure their data into files, how these data files are to be organised and how to associate metadata with these data files (*Hart et al., 2016*; *Wickham, 2014*; *Leek, 2018*).

Corresponding authors
Tjelvar S.G. Olsson,
tjelvar.olsson@jic.ac.uk
Matthew Hartley,
matthew.hartley@jic.ac.uk

Traditional scientific data management consists of individual researchers recording observations in laboratory notebooks. At another end of the spectrum, there are organisations dedicated to curating and hosting scientific data, examples from our field (biology) include the EBI (*Cook et al., 2018*), UniProt (*The UniProt Consortium, 2017*) and the Sequence Read Archive (*Leinonen, Sugawara & Shumway, 2011*).

In between these two solutions, there is a variety of systems aimed at simplifying data management for particular types of data. Laboratory Information Management Systems provide ways to manage and categorise certain types of data. Traditionally these were oriented towards sample management and they often rely on central databases. More specialised systems for managing data produced by certain types of instruments also exist. OMERO (*Allan et al., 2012*), for example, is a system aimed at managing microscopy data. These systems also tend to rely on central databases.

More generic solutions for managing data exist. One example is iRODS (*Rajasekar et al., 2015*), which provides the ability to build up capacious storage solutions by allowing access to distributed storage assets and associating data items with metadata stored in a central database. Another example is openBIS (*Bauch et al., 2011*), a framework for constructing information systems for managing biological data. OpenBIS is similar to iRODS in that it is a hybrid data repository with metadata stored in a database for fast querying and data as flat files. These systems are flexible, but require effort to customise (*Chiang et al., 2011*).

A generic format for structuring data without the need for a central database is BagIt (*Kunze et al., 2018*). BagIt is a set of hierarchical file layout conventions designed to make data storage and transfer safer. It is used in digital curation, particularly within libraries.

However, despite all these offerings there is a lack of tools for basic data management at the level of research institutes, research groups and individual researchers.

## Our motivations

Our data management challenges occur at the John Innes Centre (JIC), an independent research institute in plant and microbial sciences. Like many research institutions, the JIC has a strongly decentralised structure and culture. The 40+ research groups act mostly as independent units.

This poses a significant challenge to any data management process, and renders many existing solutions, which rely on enforced compliance with centralised systems, difficult to use.

This also tends to lead to situations where key metadata are encoded in file names and paths, metadata which can easily be lost when files are moved. Coupled with concern about actual data loss/corruption when moving files, this leads to reluctance to archive data. As a result, storage systems become full and data accumulates.

The JIC has a mixture of different storage technologies bought at different times. Each technology has its own quirks with which the end user needs to gain familiarity. For example, access to data on a file system and object-based storage work differently, user and group management varies from system to system and some technologies provide the ability to access deleted files while others do not. Having to manage different storage systems is not a productive use of researcher's time.

We needed a solution that would:

- Provide clear, immediate benefit to the front line data managers, such as core facility staff or bioinformaticians embedded in research groups.
- Allow group leaders and institute management to get an overview of the data they have.
- Enable use of different storage systems and technologies, without changing tools and pipelines.

The solution needed to be easy to use to get users started without a long learning process. It also needed to avoid lengthy migration of data to centralised platforms.

These are all common requirements for those managing data in heterogeneous research environments. Therefore any solution that meets these needs is likely to be valuable to a wide range of researchers and support groups, particularly those without existing centralised data management systems.

Here we describe our lightweight approach to managing data. It centers around the concept of packaging metadata with data, and working with the two as a unified, portable whole.

## METHODOLOGY

### Packaging data and metadata into a unified whole

Our solution to this data management problem is dtool. It is lightweight in that it has no requirements for a (central) database. It simply consists of a command line tool and an application programming interface (API) for packaging and interacting with data.

The most important aspect of dtool is that it packages data files with accompanying metadata into a unified whole. The packaged data and metadata is referred to as a dataset. Having the metadata associated with the data means that datasets can be moved around and organised. It also means that the dataset contains all the information required to verify the integrity of the data within it.

To illustrate the benefits of packaging data and associated metadata into a unified whole, it is worth comparing it to other alternatives. A common solution is to store metadata in file names and directory structures. For example consider a file named `col0_chitin_leaf_1.tif` stored in a directory named `repl_2`. The file name contains several pieces of metadata, namely that the image is of leaf sample 1 (`leaf_1`), of the Colombia-0 ecotype of *A. thaliana* (`col0`), treated with chitin (`chitin`). Furthermore the information that this is replicate 2 (`repl_2`) is encoded in the directory structure. This makes it hard to move this data around without accidentally losing metadata, since this metadata is implicitly encoded in the file path, rendering this approach fragile.

Another common approach is to store metadata in a database, this is the solution used by systems such as iRODS and openBIS. A database is quite a heavyweight solution for managing metadata. It has the disadvantage that one needs access to the database to be able to work with the data, making it difficult to work off-site when the database is managed centrally within an institute. It also makes it difficult to move data into other systems.
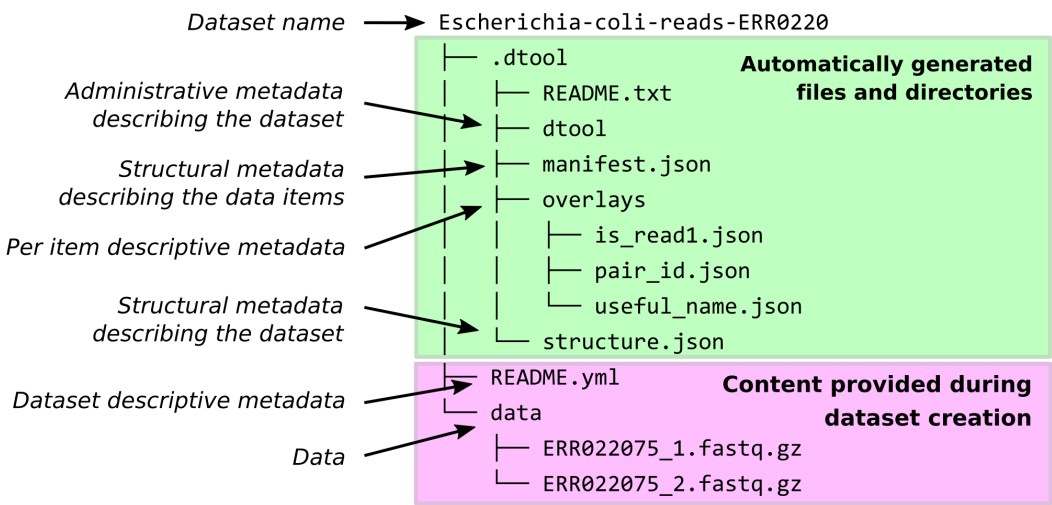

**Figure 1** **Dataset structure.** Dataset representation on file system. The user provides the data and the descriptive metadata. The software generates the administrative and the structural metadata. Per item descriptive metadata can be added to datasets as overlays. This is usually done by writing a script to annotate the items in the dataset. The script `create_paired_read_overlays_from_fname.py` in the supplementary material is an example of such a script.

When dtool is used to create a dataset, it generates both administrative metadata and structural metadata (Fig. 1). The administrative metadata contains information that helps manage the dataset and includes, for example, an automatically generated universally unique identifier (UUID). The structural metadata describes how the dataset is put together, for example each data item in the dataset has associated information about its size in a manifest, stored as part of the dataset.

When creating a dataset the user is asked to add descriptive metadata about the dataset (Fig. 2). The user is, for example, prompted to describe the dataset, state the project name and whether or not the dataset contains any confidential or personally identifiable information.

### Design decisions

The structure of the dataset was designed to be able to outlive the tool used to generate it. In practise this means that metadata files are plain text and make use of standard file formats such as JSON and YAML. It also means that there are files dedicated to describing the structure of the dataset itself (`.dtool/README.txt` and `.dtool/structure.json` in Fig. 1).

The structural metadata in the manifest (`.dtool/manifest.json`) was designed to be able to verify the content of a dataset and abstract away file paths. It therefore stores the size, checksum, and relative path of each item in the dataset. Below is the content of a sample manifest.

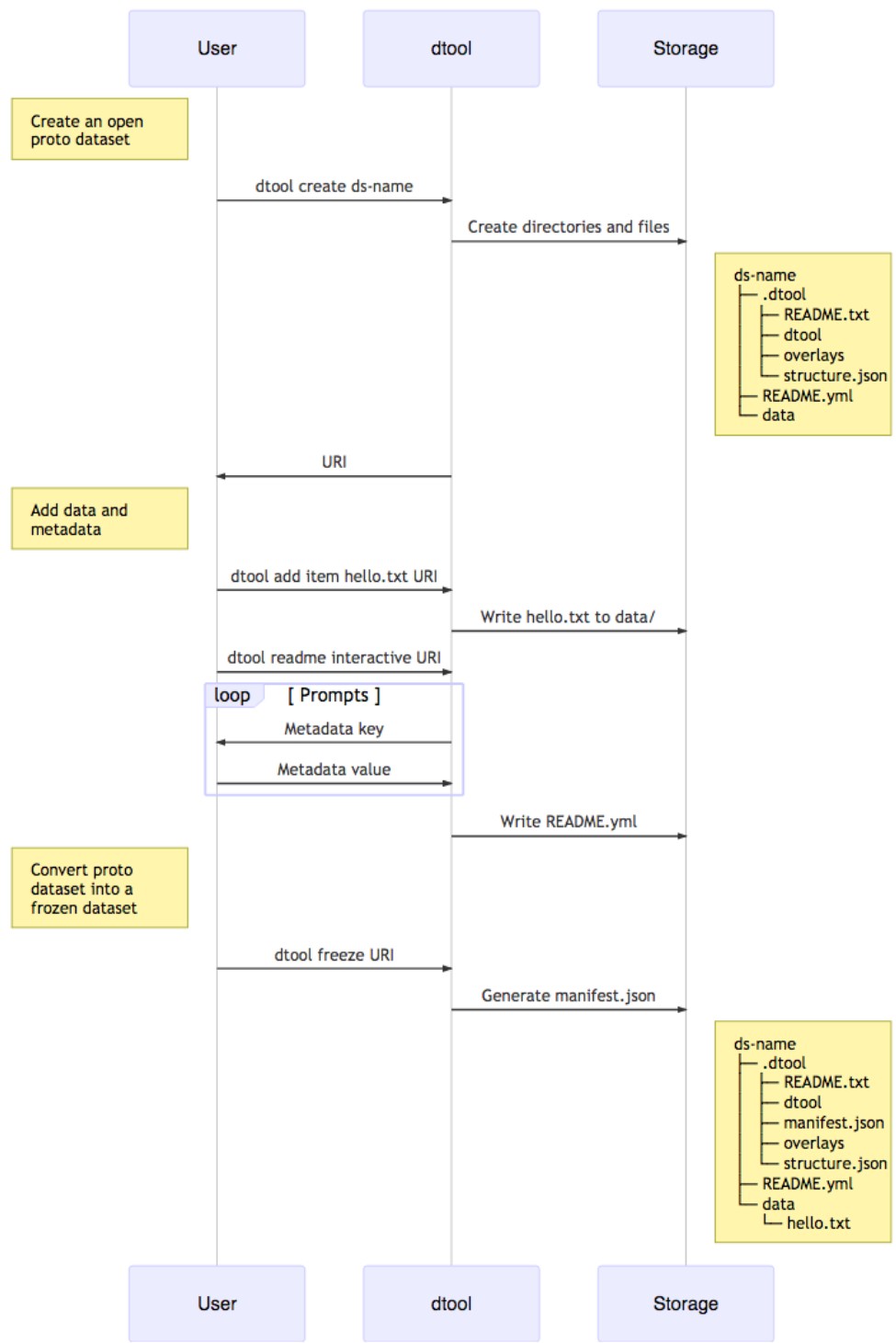

**Figure 2 Dataset creation.** The interactions between the end user, dtool and the underlying storage illustrated using a sequence diagram. In this example the dataset is given the name ds-name and it has one file added to it hello.txt. The dataset structure is illustrated as it would appear in a file system. The end user interaction with the dataset using dtool is the same for datasets stored in file system and datasets stored in object storage even though their underlying representations in the storage systems may be different.

```
{
  "dtoolcore_version": "3.7.0",
  "hash_function": "md5sum_hexdigest",
  "items": {
    "8bda245a8cd526673aab775f90206c8b67d196af": {
      "hash": "78608288625e084d68b4b5e8c1725933",
      "relpath": "ERR022075_2.fastq.gz",
      "size_in_bytes": 1982742623,
      "utc_timestamp": 1537805410.0
    },
    "9760280dc6313d3bb598fa03c5931a7f037d7ffc": {
      "hash": "4766e60e72da987e4c54e47fee5653a4",
      "relpath": "ERR022075_1.fastq.gz",
      "size_in_bytes": 1875702420,
      "utc_timestamp": 1537805358.0
    }
  }
}
```

dtool was designed to abstract away the underlying storage system. The structure of a dataset on a file system is illustrated in Fig. 1. The representation in other storage systems, such as object storage, can be different. However, the end user's interactions with dtool remain the same (Fig. 2), no matter what the underlying storage system is.

To ensure stability, the core code base (`dtoolcore`) has no dependencies outside of the Python standard library. The core code base has a high test coverage (91%) to minimise the chance of accidentally introducing bugs when developing new features. The high test coverage is a result of insisting on tests for all core code that relate to the creation and interaction with datasets. Tests were implemented using the `pytest` package and coverage was measured using CodeCov.

The dtool storage backends and the command line interface make use of a pluggable architecture, which makes it possible to write separate Python packages for creating custom storage backends and adding new sub-commands to the dtool command line interface without touching the existing code bases.

## Language choice

The dtool API and command line tool were implemented in Python. Python was chosen as it is open source, freely available and well established in the scientific community with a rich ecosystem of scientific packages. It was also chosen because it is the language the authors prefer to program in.

## Source code and documentation

The dtool source code is freely available under the liberal MIT licence on GitHub. The architecture of the code is pluggable with a core available at https://github.com/jic-dtool/dtoolcore.

The dtool documentation was created using Sphinx and is hosted using Read the Docs. It is available at http://dtool.readthedocs.io/.

Example scripts in the Results section and the supplementary material are available at https://github.com/jic-dtool/dtool_examples.

## RESULTS

From an end user's point of view there are several benefits to making use of dtool. First and foremost it provides a means to make data understandable in the future.

This is achieved by providing a standardised way to annotate a dataset with descriptive metadata.

dtool also enables researchers to:

- Back up raw data and archive old data.
- Safely move data from expensive to more cost effective storage solutions.

Both of the above are achieved by abstracting away file paths and storage technologies from the end user. In other words interacting with a dataset stored in the cloud feels the same as interacting with a dataset stored on local disk.

The abstraction of file paths and storage technologies also provides a more subtle benefit. It enables end users to write processing scripts that are agnostic to the data location, making processing scripts more portable and re-usable.

The ability to upload and download datasets to cloud storage solutions also provides the benefit of enabling researchers to share datasets with collaborators.

### Use case: making sense of data

One of the challenges in starting work in a new lab is coming to grips with old lab members' data. The person who generated the data is often no longer around and substantial effort can be spent trying to understand the context of the data and the way it has been structured.

dtool enables researchers to understand the context and content of a dataset by packaging the metadata with the data. In other words one can quickly get from a URI specifying the location of a dataset to an overview of the dataset. The URL below represents a dataset hosted in Amazon S3 that can be accessed by anyone using dtool (the URL is not intended to be displayed using a web browser).

```
http://bit.ly/Ecoli-reads
```

To find out the name of this dataset one can use the `dtool name` command.

```
$ dtool name http://bit.ly/Ecoli-reads
Escherichia-coli-reads-ERR022075
```

To find out the UUID of this dataset one can use the `dtool uuid` command.

```
$ dtool uuid http://bit.ly/Ecoli-reads
faa44606-cb86-4877-b9ea-643a3777e021
```

In the examples above dtool pulls out the name/UUID of the dataset and prints it to the terminal.

To get more information about this dataset one can use the `dtool readme show` command.

```
$ dtool readme show http://bit.ly/Ecoli-reads
---
description: Whole Genome Sequencing of Escherichia coli str. K-12 MG1655
design: Paired-end sequencing (2x100 base) of E. coli library
sample: E. coli K-12 strain MG1655
study: |
  Paired-end sequencing of the genome of Escherichia coli K-12
  strain MG1655
  using the Illumina Genome Analyzer IIx
Library:
  Name: CT1093
  Instrument: Illumina Genome Analyzer IIx
  Strategy: WGS
  Source: GENOMIC
  Selection: RANDOM
  Layout: PAIRED
  Construction protocol: |
    Standard Illumina paired-end library construction protocol.
    Genomic DNA was
    randomly fragmented using nebulisation and a ~600 bp fraction
    (including adapters) was obtained by gel electrophoresis.
links:
  - SRA: https://www.ncbi.nlm.nih.gov/sra/ERX008638
  - ENA: https://www.ebi.ac.uk/ena/data/view/ERX008638
```

The command above pulls out the descriptive metadata from the dataset and prints it to the terminal. In this case the descriptive metadata tells us, amongst other things, that this dataset contains paired-end sequencing data for *E. coli* K-12 strain MG1655.

To see the size of the dataset one can use the `dtool summary` command.

```
$ dtool summary http://bit.ly/Ecoli-reads
name: Escherichia-coli-reads-ERR022075
uuid: faa44606-cb86-4877-b9ea-643a3777e021
creator_username: olssont
number_of_items: 2
size: 3.6GiB
frozen_at: 2018-09-26
```

This reveals that the dataset contains two items and is 3.6GiB in size. The items in the dataset can be listed using the `dtool ls` command.

```
$ dtool ls --verbose http://bit.ly/Ecoli-reads
8bda245a8cd526673aab775f90206c8b67d196af   1.8GiB  ERR022075_2.fastq.gz
9760280dc6313d3bb598fa03c5931a7f037d7ffc   1.7GiB  ERR022075_1.fastq.gz
```

In the above the `-v/--verbose` flag is used to return the size as well as the identifier and the relative path of each item.

In summary the commands `dtool readme show`, `dtool summary` and `dtool ls` give an overview of the context and content of a dataset.

## Use case: backing up raw data and archiving old data

At the JIC we have several storage solutions, each one serving a specific purpose. Relatively expensive file system storage is used for processing data. Object storage accessible via the S3 API, with off-site backups is used for storing raw data. A capacious storage system front-ended by iRODS is used for archiving long term intermediate data. Because dtool abstracts away the underlying storage solution the end users can use the same commands for copying data to and from these differing storage systems. The ease of moving data around can be illustrated by copying a dataset hosted in the cloud to local disk.

```
$ dtool cp -q http://bit.ly/Ecoli-ref-genome .
file:///Users/olssont/Escherichia-coli-ref-genome
```

In the above the `-q/--quiet` flag is used to only return the URI specifying the location that the dataset has been copied to, in this case a directory named `Escherichia-coli-ref-genome` in the current working directory.

dtool can be used to copy a dataset between different storage solutions. This enables researchers to copy data to storage solutions designed for backing up and archiving data.

## Use case: generating inventories of datasets

One of the challenges of running a research group is keeping track of all the data being generated. As such it is useful to be able to list datasets and to generate inventories of datasets. This can be achieved using the commands `dtool ls` and `dtool inventory`.

The purpose of `dtool ls` is to provide a way to list names and URIs of datasets. Below is an example of the `dtool ls` command listing three datasets stored in a directory named `my_datasets` (see the supplementary material for details on how to setup this directory).

```
$ dtool ls my_datasets
Escherichia-coli-reads-ERR022075
  file:///Users/olssont/my_datasets/Escherichia-coli-reads-ERR022075
Escherichia-coli-reads-ERR022075-minified
  file:///Users/olssont/my_datasets/Escherichia-coli-reads-
ERR022075-minified
Escherichia-coli-ref-genome
  file:///Users/olssont/my_datasets/Escherichia-coli-ref-genome
```

The need for this command becomes more apparent when working with datasets stored in the cloud. The command below lists datasets in the Amazon S3 bucket `dtool-demo`.

Note that the command below requires the user to have permissions to read the bucket and as such will not work for the readers of the paper, but is included for illustrative purposes.

```
$ dtool ls s3://dtool-demo/
Escherichia-coli-ref-genome
  s3://dtool-demo/8ecd8e05-558a-48e2-b563-0c9ea273e71e
Escherichia-coli-reads-ERR022075-minified
  s3://dtool-demo/907e1b52-d649-476a-b0bc-643ef769a7d9
Escherichia-coli-reads-ERR022075
  s3://dtool-demo/faa44606-cb86-4877-b9ea-643a3777e021
```

The `dtool inventory` command is intended to be able to generate reports of datasets. The command below creates a report (`my_datasets.html`) listing all the datasets in the `my_datasets` directory as a single HTML file that can be shared with colleagues.

```
$ dtool inventory --format=html my_datasets > my_datasets.html
```

In summary the `dtool ls` command can be used to find data in a base URI and `dtool inventory` can be used to generate reports and web pages to make datasets findable.

## Use case: verifying the integrity of old data

It is useful for researchers to be able to reassure themselves that their research data is intact.

In order to be able to check whether or not this is the case, dtool provides a means to verify the integrity of a dataset, using the `dtool verify` command. By default this checks that the expected item identifiers are present in the dataset and that files have the correct size. There is also a further option to check that the content of the files is correct by comparing checksums, using the `-f/--full` option.

```
$ dtool verify Escherichia-coli-ref-genome
All good :)
```

The command above shows that the dataset contains the expected content. To illustrate what happens if a dataset becomes corrupted we can move a file out of the dataset.

```
$ mv Escherichia-coli-ref-genome/data/U00096.3.fasta .
$ dtool verify Escherichia-coli-ref-genome
Missing item: b445ff5a1e468ab48628a00a944cac2e007fb9bc U00096.3.fasta
```

In summary dtool provides a means to get clarity with regards to the integrity of a dataset. This is not, however, intended to detect deliberate tampering with the dataset, i.e., it would be possible to make the `dtool verify` command pass by deliberate modification of the underlying files and the manifest.

## Use case: processing data

dtool provides programmatic access to the data in a dataset. This means that one can use dtool to create scripts that abstract away the location of the data.

For example, to process all the items in a dataset one can use the `dtool identifiers` command to list all the identifiers. To access the content of the items one can then use the `dtool item fetch` command, which returns the absolute path to a location where the item can be read. For datasets stored in the cloud, the `dtool item fetch` implicitly includes a step to download the item to local disk to ensure it can be read from the absolute path returned by the command.

Below is a Bash script (`simple_processing.sh`) to illustrate this. The script extracts the first line from each dataset item, using `gunzip` and `head`.

```bash
#!/bin/bash

# Read in the input dataset URI from the command line.
INPUT_DS_URI=$1

# Process all the items in the input dataset.
for ITEM_ID in `dtool identifiers $INPUT_DS_URI`; do
    # Fetch an item and process it.
    ITEM_ABSPATH=`dtool item fetch $INPUT_DS_URI $ITEM_ID`
    gunzip -c $ITEM_ABSPATH | head -n 1
done
```

Running this `simple_procssing.sh` script on a dataset stored in the cloud is illustrated below.

```
$ bash simple_processing.sh https://bit.ly/Ecoli-reads-minified
@ERR022075.1 EAS600_70:5:1:1158:949/2
@ERR022075.1 EAS600_70:5:1:1158:949/1
```

We can verify that this gives the same results as running the script on a dataset stored on local disk by copying the dataset and re-running the script on the local dataset.

```
$ LOCAL_DS_URI=`dtool cp -q https://bit.ly/Ecoli-reads-minified .`
$ bash simple_processing.sh $LOCAL_DS_URI
@ERR022075.1 EAS600_70:5:1:1158:949/2
@ERR022075.1 EAS600_70:5:1:1158:949/1
```

It is also possible to use dtool to store the output of processing scripts, both in terms of data and metadata. In other words, it is possible to implement scripts that implement dataset to dataset processing. This can be used to automate aspects of data management.

The script below, called `minfiy.sh`, uses this concept of dataset to dataset processing. It is worth noting that the script:

- Creates a name for the output dataset based on the input dataset name
- Creates an output dataset
- Processes all the items from the input dataset and adds the results to the output dataset
- Extracts the descriptive metadata from the input dataset as a base for the descriptive metadata of the output dataset

- Adds a reference to the input dataset and a description of how it was processed to the descriptive metadata of the output dataset

```
#!/bin/bash

# Exit immediately on failure of a command.
set -e

# Read in the input from the command line.
INPUT_URI=$1
OUTPUT_BASE_URI=$2
NUM_LINES=4000

# Create a name for the output dataset based on the input dataset.
OUTPUT_NAME=`dtool name $INPUT_URI`-minified

# Create an open proto dataset.
OUTPUT_URI=`dtool create -q $OUTPUT_NAME $OUTPUT_BASE_URI`

# Process all the items in the input dataset and
# add the results to the output dataset.
for ITEM_ID in `dtool identifiers $INPUT_URI`; do

  # Fetch the item from the dataset and get an absolute path
  # from where its content can be accessed.
  ITEM_ABSPATH=`dtool item fetch $INPUT_URI $ITEM_ID`

  # Write the minified version of the item to a temporary file.
  TMP_MINIFIED=$(mktemp /tmp/minfied.XXXXXX)
  gunzip -c $ITEM_ABSPATH | head -n $NUM_LINES | gzip > $TMP_MINIFIED

  # Add the temporary file to the output dataset giving it the relpath
  # of the item from the input dataset.
  RELPATH=`dtool item relpath $INPUT_URI $ITEM_ID`
  dtool add item $TMP_MINIFIED $OUTPUT_URI $RELPATH

  # Cleanup.
  rm $TMP_MINIFIED
done

# Create descriptive metadata for the output dataset.
TMP_README=$(mktemp /tmp/dtool-readme.XXXXXX)
dtool readme show $INPUT_URI > $TMP_README
```

```
echo "minified:" >> $TMP_README
echo "  from_UUID: 'dtool uuid $INPUT_URI'" >> $TMP_README
echo "  from_URI: $INPUT_URI" >> $TMP_README
echo "  content: first $NUM_LINES per item" >> $TMP_README

# Add the descriptive metadata to the output dataset.
dtool readme write $OUTPUT_URI $TMP_README

# Cleanup.
rm $TMP_README

# Finalise the output dataset.
dtool freeze $OUTPUT_URI
```

In Supplemental Information 1 there is a script that performs a Bowtie2 (*Langmead & Salzberg, 2012*) alignment. It takes as input a dataset with paired RNA sequencing reads, a dataset with a reference genome and a base URI specifying where the output dataset should be written to. The command below shows the usage of this script.

```
$ bash bowtie2_align.sh  \
  http://bit.ly/Ecoli-reads-minified  \
  http://bit.ly/Ecoli-ref-genome .
```

Running this command creates a dataset in the current working directory. Below is a command to create an environment variable as a simple alias to this directory.

```
$ DS_URI=Escherichia-coli-reads-ERR022075-minified-bowtie2-align
```

The content of this dataset is a SAM (Sequence Alignment Map) file.

```
$ dtool ls $DS_URI
3ffaeaf15fc1f12417aadddb9617fb048e39509e  ERR022075.sam
```

The descriptive metadata gives information about how this SAM file was derived.

```
$ dtool readme show $DS_URI
---
description: bowtie2 alignment
input_reads_uri: http://bit.ly/Ecoli-reads-minified
ref_genome_uri: http://bit.ly/Ecoli-ref-genome
bowtie_version: bowtie2-align-s version 2.3.3
```

It is important to note that the metadata above was generated automatically by the `bowtie2_align.sh` script.

In summary, dtool provides a means to write processing scripts that are agnostic to where the input data is stored, whether it be on local disk or in some object storage system in the cloud. Furthemore, using dtool to store the data generated from processing scripts allow researchers to automate parts of their data management.

## Use case: sharing data

It is possible to share datasets hosted in cloud storage such as Amazon S3 and Microsoft Azure storage.

Take for example the dataset represented by the URI below.

```
s3://dtool-demo/8ecd8e05-558a-48e2-b563-0c9ea273e71e
```

This is the dataset with the *E. coli* reference genome data.

```
$ dtool name s3://dtool-demo/8ecd8e05-558a-48e2-b563-0c9ea273e71e
Escherichia-coli-ref-genome
```

This URI can only be used by people that have been authorised to interact with the dtool-demo Amazon S3 bucket. To make this dataset accessible to the public one can use the `dtool publish` command.

```
$ dtool publish -q  \
  s3://dtool-demo/8ecd8e05-558a-48e2-b563-0c9ea273e71e
https://dtool-demo.s3.amazonaws.com/8ecd8e05-558a-48e2-b563-0c9ea273e71e
```

It is now possible for anyone in the world to interact with this dataset using the HTTPS URI returned by the `dtool publish` command.

```
$ dtool name  \
  https://dtool-demo.s3.amazonaws.com/8ecd8e05-558a-48e2-b563-0c9ea273e71e
Escherichia-coli-ref-genome
```

To make life easier one can use a URL shortner like Bit.ly to create a more user friendly URI. The example below refers to the same dataset as above.

```
dtool name http://bit.ly/Ecoli-ref-genome
Escherichia-coli-ref-genome
```

In summary dtool makes it possible to share datasets with collaborators and to make datasets accessible to the research community.

## Uptake at JIC

At the JIC we have introduced twelve research groups that deal with high volumes of data (>10TB) to dtool. Of these, five groups have adopted dtool to manage their data, another four are still evaluating it and three have found it difficult to move away from their existing working practises.

Of the three groups that have found it difficult to move away from their exising working practises the main barriers to uptake have included: (1) feeling overwhelmed with the amount of exising data; (2) tight coupling of data analysis and input files, i.e., hard coded paths in scripts; (3) use of directory structures to encode provenance of data analysis; (4) absence of a data management champion within the group.

Four of the five groups that have adopted dtool found the incentive to do so in association with a key member leaving the group. The other group that adopted dtool did so as part of

starting a new project. In this instance we helped the group devise a protocol for packaging raw sequencing data into datasets as it arrived from the sequencing company.

To date 26TB of data has been packaged into 803 datasets.

## DISCUSSION

Data management provides a set of difficult problems. Technological developments in scientific instruments, such as high throughput sequencers and super resolution microscopes, have led to an explosion of data that must be stored, processed and shared. This requires both recording of appropriate metadata and ensuring consistency of data.

These problems are compounded by the issue that those directly generating and handling data, often junior researchers, have different immediate incentives from funders and institutions. These front-line researchers need to be able to quickly receive and process their data to generate scientific insights, without investing substantial time learning to use complex data management systems.

However, maintenance and sharing of data is critical for the long term success of science. Funding bodies therefore put requirements on grant holders to ensure that the data generated by projects are shared and discoverable.

While there are good theoretical guidelines for data management, there is a lack of tools to support them, particularly in the decentralised environment in which research takes place.

Our attempts to solve these challenges led us to develop dtool. This tool provides a quick and straightforward way to package a collection of related files together with key metadata, which we term a dataset. This dataset provides both consistency checking and access to both dataset and file level metadata, while being portable.

During the development of dtool we were not aware of the BagIt (*Kunze et al., 2018*) file packaging format. Despite this, during its development, dtool evolved to share several features with BagIt. In particular, dtool's disk storage implementation also makes use of file manifests containing checksums, descriptive metadata in plain text format, and a flexible system for annotation with further metadata.

dtool differs, however, in that it is fundamentally an API that provides an abstract interface to data and metadata, while BagIt is a specific way in which to store data on a standard file system. dtool provides the ability to store data and metadata on different storage systems (file system disk, iRODS, S3, and Azure at present, with a structure that allows further extension). This is coupled with provision of a consistent interface for working with this data that is independent of the storage medium. These different natures give rise to different use cases—BagIt for ensuring safety of data in transit and at rest, dtool to enable programmatic data management processes and use of different storage technologies.

Beyond this difference in purpose (format versus abstraction tool), dtool provides the ability to programmatically set and work with per-item metadata through its overlay system. This could be achieved using BagIt's tag system, by implementing a consistent convention for storage of this metadata on top of BagIt.

In future work we would like to experiment with creating a custom storage broker for writing datasets to disk following the BagIt file packaging format. If successful, this could pave the way to making it dtool's default on disk packing format.

dtool datasets have been designed in accordance with the principles for storing digital data outlined in *Hart et al. (2016)*. dtool leaves original files intact and uses markup to add additional metadata, adhering to the principle of keeping raw data raw. The markup used by dtool is plain text files using standard formats such as YAML and JSON, in line with the principle of storing data in open formats. Each dtool dataset is given a UUID and each item in a dataset has a unique identifier, thus meeting the principle that data should be uniquely identifiable.

dtool datasets are a good fit with many of the ideas regarding the life cycle of data (*Michener, 2015*). The life cycle of data centres around the concepts of defining how data will be organised, documented, stored and disseminated. By making it easy to move datasets around, dtool provides a solution for the organisation, storage and dissemination of data. By allowing the metadata to be packaged alongside the data, dtool also provides a solution for documenting data.

dtool also provides a means for individual researchers and research groups to share their data using cloud hosting solutions in a manner that makes the data adhere closely to the FAIR principles (*Wilkinson et al., 2016*). While cloud hosting requires a consistent funding stream, dtool provides the technical capabilities to facilitate Accessibility, Interoperability, and Resusability. The core Python API provides programmatic access to metadata that can be used to create tools for indexing and searching the datasets. Therefore it provides the potential to make the data Findable.

dtool has provided substantial benefits for the research groups at JIC that have started using it. Dataset consistency checking has given the researchers peace of mind that the key data underpinning their scientific results are safe and secure. Requiring entry of appropriate metadata when datasets are created has led to better organisation of data and the ability to retrieve and understand data after capture and storage. The ability of the tool to easily move datasets between the different storage systems to which they have access has substantially reduced their storage costs, translating into increased capacity to store and process data with the same resources.

However, there are hurdles to the uptake of dtool. Particularly when people feel overwhelmed by the amount of data they have. We have found that for a research group to start making use of dtool they need some external incentive. To date, the most common one has been a key member leaving the group. The incentive for the groups that are still evaluating dtool is to free up working space on expensive compute cluster associated storage. The latter seems to have less weight than the former as no group has yet started using dtool in order to do the latter.

Another hurdle to the uptake of dtool has been that it is a command line tool. This means that some group leaders, who are less unfamiliar with the command line, have some reservations about the tool. We are currently working on building graphical tools to overcome this.

We did not initially plan on deployment to systematically collect information on usage, strengths and barriers to using dtool. As we continue to use and deploy dtool at JIC, we will more systematically collect information and work with users to continue to improve dtool.

dtool is now a key part of the JIC's data management strategy. This means that members of the core funded JIC Informatics team will maintain the tool and expand the software eco-system surrounding it.

## CONCLUSION

Without data management, reproducible science is impossible. Our rapidly expanding ability to collect and process data has the potential to generate important insights. However, this is only possible if the data is accessible and the person doing the analysis has enough knowledge about the observations in the data to put them into the context of a research question. Making data accessible and understandable becomes increasingly complex as the volumes of data grow.

In particular, there are substantial challenges in: capturing and storing metadata together with data; ensuring consistency of data that is comprised of multiple individual files; and being able to use heterogeneous storage systems with different capabilities and access methods. These challenges become even more difficult to overcome in the highly decentralised environment in which much scientific research takes place.

dtool helps researchers overcome these challenges by providing a lightweight and flexible way to package individual files and metadata into a unified whole, which we term a dataset. This dataset provides consistency checking, giving reseachers confidence that their data maintains integrity while moving it between storage systems. Storing key file- and dataset-level metadata together with the data allows the data to be understood in future. The ability to use different storage backends such as file system, iRODS, S3 or Azure storage allows data to be moved to the most appropriate location to balance cost and accessibility.

dtool datasets are ideally suited for processing because one can access both the data and the metadata programatically. Further, it is possible to create new datasets for storing the output of processing pipelines programatically using both the command line tool and the Python API. This provides a means to automate aspects of data management by incorporating it into processing pipelines.

Our tool is available as free open source software under the MIT license. We hope that it will provide benefit to others.

## ACKNOWLEDGEMENTS

We thank Adi Kliot and Hugh Woolfenden for critical reading and feedback on an early version of the manuscript. We thank the bioinformaticians at the John Innes Centre that have been kind enough to test and provide feedback on dtool. We thank the PeerJ editor and the reviewers for their constructive feedback that resulted in a much improved paper and enhancements to the dtool software.

### Funding

This research was supported by the Biotechnology and Biological Sciences Research Council. The funders had no role in study design, data collection and analysis, decision to publish, or preparation of the manuscript.

### Grant Disclosures

The following grant information was disclosed by the authors:
Biotechnology and Biological Sciences Research Council.

### Competing Interests

The authors declare there are no competing interests.

### Author Contributions

- Tjelvar S.G. Olsson conceived and designed the experiments, performed the experiments, analyzed the data, contributed reagents/materials/analysis tools, prepared figures and/or tables, authored or reviewed drafts of the paper, approved the final draft.
- Matthew Hartley conceived and designed the experiments, performed the experiments, analyzed the data, contributed reagents/materials/analysis tools, authored or reviewed drafts of the paper, approved the final draft.

### Data Availability

The core code is available at:

https://github.com/jic-dtool/dtoolcore

The dtool command line application is available at:

https://github.com/jic-dtool/dtool

Raw data are available at:

https://github.com/BenLangmead/bowtie2/tree/80edefea19084d5b027a46f2e4feaae949d6a11c/example/reads

### Supplemental Information

Supplemental information for this article can be found online at http://dx.doi.org/10.7717/peerj.6562#supplemental-information.

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
