# Peer review of "Lightweight data management with dtool"

_PeerJ, doi:10.7717/peerj.6562_

## Round 0.1 · original submission · Major Revisions

Particular weight should be given to the comments regarding the appropriateness of the figures, the organization of the sections, and the places where statements and conclusions are not backed by scholarship or evidence.

·

Basic reporting

> _Clear, unambiguous, professional English language used throughout._

The manuscript is written clearly and understandably.

> _Intro & background to show context. Literature well referenced & relevant._

The introduction includes a reasonable description of the problem that the software aims to address.

However, some important and seemingly relevant prior art and corresponding work is not mentioned at all. In particular, one could get the impression that the presented data packaging format and structure is in essence a reinvention of the BagIt standard (<https://en.wikipedia.org/wiki/BagIt>). There may be reasons why this is not quite the case, but the relevancy of BagIt seems strong enough that it should at least be discussed how this differs, and why the authors chose to create their own standards instead of adopting BagIt as an existing standard for package structure and metadata.

> _Structure conforms to PeerJ standards, discipline norm, or improved for clarity._

The manuscript follows its own, alternative, structure. I'm not convinced that the result improves clarity substantially over a more standard structure. Although the chosen structure lends itself well to a descriptive story about how the software can be useful, a more critical reflection on limitations and their underlying design choices is missing. Also, the chosen format doesn't seem to leave no obvious place for putting observational data about data management practices with and without the software (whether as the result of an experiment or anecdotal).

I'd suggest to prefix the subheadings under _Extended use cases_ with "Use case:" or something similar. Or perhaps first give an enumeration of the extended use cases being described? Or both? In the present form, it's easy to lose track for a section that it presents one of the extended use cases.

The fact that there is a section _Extended use cases_ implies that there were also standard (? basic? original?) use cases. But the text does not make it clear what those were, and hence it is unclear how the distinction between extended and other use cases is meaningful. (The "extended" use cases described here all look pretty basic to me for a generic tool aimed at addressing the most common data management challenges.)


> _Figures are relevant, high quality, well labelled & described._

Figures are relevant and good quality, but they are also simply cartoons. This may be exactly right for a software manual, but seems overly simplistic for a scientific article. In other words, by confining themselves to these high-level cartoons, I think the authors are completely missing the opportunity to actually visually present specifics about their software and data package design choices. For example, the structure of the package, including the structure of the metadata directory (`.dtool/`) and its contents within the package are almost entirely missing from the manuscript. (Notice that even the Wikipedia article on BagIt does include this.) Sequence, use case, or activity diagrams (whether UML style or not) are also missing but could be used to present the design of user to software, state to state, and software to storage interactions.

> _Raw data supplied (see PeerJ policy)._

Source code (aside from source code of the presented software) for various scripts and tasks is provided, but printed in the manuscript and in the supplementary file. That requires the reader to copy and paste out of the text, which is needlessly laborious and error prone. Why not provide these as text files, whether under version control (in a Github repo or as Gists), or simply deposited in an archive for research products (such as Zenodo).

Experimental design

> _Original primary research within Scope of the journal._

Yes.

> _Research question well defined, relevant & meaningful. It is stated how the research fills an identified knowledge gap._

Yes.

> _Rigorous investigation performed to a high technical & ethical standard._
> _Methods described with sufficient detail & information to replicate._

There aren't experiments and their results being reported here to which this would apply.

Validity of the findings

The manuscript reports a positive impact of the presented software on data management-related practices at the author's home institution. However, no experiments, metrics, specific observations, or any other data are presented to support this finding. I suspect the authors never collected any before-after data that coul support such findings. However, this is not a minor question. Presumably, creating the software wasn't the objective as an end onto itself, but was expected to be a means towards an end. Hence, simply being silent about the observed impact on data management practices because there are no hard data seems rather unsatisfactory too. If the authors are confident about the positive impact, perhaps they can at least find and describe some concrete cases, even if these had mostly anecdotal character.

The text also in several places speaks about FAIR data, and that the presented software would help with that. I agree that the software should help promote interoperability and reusability, though primarily through the attachment of metadata to the data package. There's also the ability to make publicly available a dataset stored in the cloud, but, given the fact that such storage is ephemeral (it will disappear as soon as one of the bucket owners stops paying the bill, or decides to delete it) isn't that about as good or bad as data stored on some lab's web server? And how does the software help with findability? So it seems how and where this software fits into promoting FAIR data is a little more nuanced as appears from the manuscript, and would benefit from a more critical discussion.

The section about verifying the integrity of a data set needs a more critical treatment of limitations. The verification as currently implemented isn't tamper-proof in any way, and even if this may not be of much significance to individual investigators, I know it is for many institutions, and also for data-producing core facilities. (See under _General comments_.)

Additional comments

Line 41: _need_ is extraneous

Line 83-87: "_there is limited immediate incentive for the people generating the data, most commonly PhD students and post-docs, to care about data management_": This is, I would argue, prone to be misunderstood, because, as the authors point out, "data management" is a broad term. Arguably there's no question that those generating or analyzing the data *have to* care about managing the data somehow. What there is less of an incentive for is to do so in such a way that the result is well-structured, and has sufficiently comprehensive metadata so that the result can be FAIR. And yes, their career relies on generating publications, but publications that receive little attention and reuse because their underlying data are inscrutable are against their self-interest. This is probably what the authors meant anyway. So I suggest the authors be specific here, and not altogether dismiss an incentive for "data management" writ large, and not generically put reusable data in opposition to academic career incentives.

Line 102: _Each technology has its own quirks with which the end user needs to gain familiarity._ I don't necessarily disagree, but on its own this reads overly opaque. Maybe supply an example how these quirks translate into adoption becoming time consuming?

Line 137: _This makes it hard to move this data around without losing metadata._ I think the problem isn't so much that it's hard to move this data around (is it really?) but that it's so *easy* to do so in a way that loses metadata without notice. I.e., isn't the problem one of vulnerability and thus high risk (of metadata loss) than of difficulty?

Line 182 and later: The S3 bucket URIs such as <http://bit.ly/Ecoli-reads> etc will look like a browser-resolvable URL to many, but yield "Access denied". To use these one will have to set up AWS credentials and use dtool (or perhaps some other AWS CLI tool?). This should probably be explained upfront to spare readers from fruitlessly clicking on these URLs (they are in fact hyperlinked -- perhaps they should not be).

Line 188, 216: It appears that `dtool readme show` returns its results in YAML format, whereas `dtool summary` returns JSON format.
Why the choices, and why not use the same format for both commands?

Line 288, "Verifying the integrity of old data". I think the far more interesting case of detecting dataset corruption, accidental or from tampering, is verifying that the file *contents* are exactly what they are expected to be. A little experiment shows that dtool doesn't detect file content corruption. Also, if one deletes a file and then also deletes the manifest entry, `dtool verify` reports that all is good. So this is just simply MANIFEST checking, with no checking of file content integrity (aside from file size checking), and no detection of tampering. Perhaps the authors consider these checks out of scope for dtool, but such intentional limitations should then be clearly stated so that the "verify" capability is not misunderstood. Cryptographic signature checking is not hard to implement and widely employed by, for example, package managers of all flavors, and hence may easily be expected to be taking place if it's not made clear that only the data manifest is being checked.

Line 328-332: The presented output from the script seems to be missing the `PROCESSING ITEM` line that the script should output, suggesting that what was run to generate the output isn't what's printed in the manuscript. Same for lines 336-339.

Line 395: This is the first time the `dtool` command `uuid` appears, which seems unfortunate and a missed opportunity?

Line 446-448: Why isn't this just another `dtool` command, such as `dtool publish`? It seems like an unnecessary surprise that doing this requires remembering a separate command line tool?

Reviewer 2 ·

Basic reporting

This paper is well written. First I have to say that I love this idea! I work in biology and bioinformatics and have been advocating for keeping data and metadata together for many years. This is a key, and under appreciated issue! I was really excited reading the paper about the potential for the tool and would be excited to try it out with my data. Before I would commit to employing it more broadly though, I would in particular want more information on the implementation of the software and to understand how it has been used and received at the institute where the authors work, and outside if others have used it. More specific comments below.

Overall it seems like this was a tool that was developed for the needs of the John Innes Centre, and this paper is the opportunity to present it to a broader audience and hope that others also want to use it and find it useful. That's great! However to make that transition, it would be helpful and important to have more of a general context for the paper throughout. In particular, that starts with the introduction. The introduction section lacks references that put the tool and paper in the context of the literature. Overall the intro doesn’t demonstrate a strong understanding or appreciation for the space. That’s potentially OK, but when presenting some new tool as better than current approaches it’s important to put it into context and also appreciate other contributions. A few particular examples are cited below.

Introduction:
- More references are needed in the introduction for such statements as "has resulted in data management becoming one of the big challenges faced by the biological sciences”. There are many reports and papers documenting this as an issue. It's a good opportunity to highlight them here, and that it is broadly recognized as an issue. Also important for recognizing others who have explored this topic.
- The statement "Open access is increasingly viewed as a public good”. I don’t think it’s ‘open access’ that’s viewed as a public good, but data, which means that data should be open access.
-Are there no more recent reference than 2015 on funding agencies requiring data sharing plans?
- I don’t think ‘laboratory notebooks’ are one extreme. They are historically and currently important.
- A few examples are cited. Why are these particular ones selected?
- In the 'Data management problem' section, overall challenges in data management are presented as ‘just so’ stories and not in relationship to research that has highlighted the challenges.
- Rather than separating ‘Data Management challenges’ and ‘Our motivation’ the authors either need to expand and add citations to the ‘Data Management’ section, or could combine into one ‘Challenges of data management at an independent research institute’. Then they could focus on their issues in particular, and also help ground the tool in the type of organizations it was developed for. The ‘Our motivation’ section was much more compelling and provided a clearer picture of why the tool would be important.

The section describing the use of the tool was exciting and made me want to try it out! Good examples and nice to include the code that would do those tasks. However, I didn't have much of a sense of how it was implemented 'under the hood' from the paper. More on that in further sections.

Similarly, while it looked exciting and useful, it would have been good to hear how it's been received where it's been used. This likely has been used at John Innes Centre. How has it been received? What have been the issues? How many people are using it? Has it been used at other places? More information on how it has been used and some ‘lessons learned’ would be very valuable. The authors have data on reliability and usage! That would be great to include and help people evaluate if they want to use it.

Experimental design

Because this is a software paper, there isn't exactly an 'experimental design' section. So, here I'll discuss 'software design'.

The authors provide links to the software and documentation. Both are excellent. I was able to install the tool and use it with the example described. Code is clear and well documented.

This paper could do with essentially a 'Methods' section that describes how the tool was implemented. What's the conceptual framework for the approach? Why was it chosen? How was it implemented? Does the implementation rely on other theoretical (e.g. graph theory, hashing) or practical implementations? This is really important for evaluating its speed and reliability and since this is a paper, is a great opportunity to discuss this component. The paper is focused on how to use the software, rather than about the software. Both are important, but if there is not a more theoretical discussion of the software, this is more of a 'how to' paper than an academic publication. I definitely recognize the challenges with software of walking this line of what to include in a paper, but I think this is important in this journal.

Validity of the findings

With a software paper, again a different model. So here I'll comment on 'usage of the tool'. As mentioned in the other paper, it's difficult to evaluate the effectiveness of the tool without information on how it has been used and even potentially 'lessons learned'. Biologists do really want to see evidence of it being used in a 'real world' scenario before they are eager to try a new approach or tool. Since the authors likely have some of this information from using it in their context, it would be very helpful to include.

This is a totally different workflow for biologists, so for adoption there is a strong social/cultural issue. For making that sort of change, this kind of information is crucial.

As this is open source and they are putting it out for usage and contributions, it would also be interesting to hear if they had plans for how they are going to sustain and support this open source project. This is not often included in papers, so the authors wouldn't' be expected to include, but it is an important element of open source and should be more discussed.

Additional comments

Thank you for this nice tool! I recognize the challenges of wanting to publish software in a journal. From my perspective, this paper needs a bit more general background, and discussion about the tool and its use for it to be included in this particular journal. That's a great goal though and would really help with adoption, so I would encourage the authors to keep working on this paper.

If they wanted to publish just the software, there are options like the Journal of Open Source Software. Publishing there would not preclude publishing here, but would get a publication and DOI for the software itself and a paper like this could focus on the background, methods and usage.

---

## Round 0.2 · Minor Revisions

I appreciate the detailed rebuttal and all the changes made in light of the earlier reviews. The reviewers make a small number of valuable suggestions on the latest version that I feel the authors should have the opportunity to address (even though they are not obstacles to acceptance). Reviewer 1 makes some suggestions for further clarifying the structure of the work and the paper, while Reviewer 2 makes some suggestion for improving the 'Uptake at JIC' section.

In addition, the revisions did not adequately address the point from Reviewer 1's comments on the earlier manuscript, which I emphasized in my decision letter, that the previous figures were problematic. The figures new to this version are good additions. But Figures 1, 2, 5 and 6 are not useful apart from their caption text and are not appropriate for a scientific publication. My decision of acceptance is conditional on the removal of the cartoon graphics (or replacement by graphics that actually convey information).

Also, a couple minor typos that I noticed:
- line 502 "26TiB"
- multiple occurrences of "(Kunze et al.)" are missing the year of publication.

·

Basic reporting

The text is substantially improved in structure and introduction of relevant prior art. The authors' responses to my earlier comments are largely satisfactory, and the introduction of the two new figures definitely add useful information to the manuscript.

That being said, two issues remain.

One, the text across the Methodology and Results section still feels in parts undecided on what in this piece of scholarship constitutes methodology, what constitutes a result, and what constitutes interpretation. For example, the Methodology section seems to consider the product that is the report's primary subject, the dtool software a methodology, in the same vein as explaining design decisions. On the other hand, major use-cases are mostly considered Results, even though presumably these drove, in the form of a-priori requirements, the need for and the design of the software from the beginning. And the description of the use-cases in the Results sometimes veers into interpreting the facts, for example where it speaks to the ease of using the tool, or that it gives a "clear overview". I.e., as described it remains unclear for this example whether ease of use and clarity of certain output is something the authors (as the developers) believe to have achieved, or something that users have reported or confirmed to them, or both. I think the text would benefit from a more thorough follow-through on the new structure. As another example, aside from an enumeration of driving requirements (which presently are sort-of implied by the use-cases in the Results), other elements of the development methodology, are glossed over as well, such as the use of Python as implementation language (was there a design reason for this choice, or is it simply what seemed up to the task and what the developers were sufficiently fluent in?), how and with which objectives (e.g., percent code coverage aimed for?) testing is implemented, and how documentation is authored, and then rendered/published.

Two, the introduction and discussion of the BagIt standard goes a long way to addressing my earlier comment about it. But it also still leaves some questions open, perhaps in part because I wasn't clear enough in my earlier comments. Specifically, the authors' point seems to be mostly that dtool is a both a tool for end-users and a programmatic interface for programmers, whereas BagIt is a standard structure for a dataset and its metadata. That is obviously both true and fair enough. However, dtool under the hood does employ a structure for a dataset and its metadata, only it is an ad-hoc created one rather than BagIt. If, as the authors state, the dataset structure was designed to survive the tool used to create it, i.e., to stand on its own, then if the authors aren't using an existing standard for this they are de-facto creating a new (a d-hoc) standard, whether they meant to or not. I think this merits a more thorough treatment in the text. The authors don't really provide a good argument as to why BagIt didn't and/or doesn't meet their needs for the underlying dataset structure. ("implementing a consistent convention for storage of this metadata on top of BagIt" seems, if possible and not overly complicated, still preferable to creating an entirely new standard ad-hoc?) Perhaps what happened is that in the design of the underlying dataset structure existing standards such as BagIt were not considered for adoption. If so, what could still be done now is to acknowledge that, and to outline steps that would need to be taken to adopt BagIt or another existing standard in a future version (such as layering some kind of metadata storage convention on top so that dtool's requirements are met). Alternatively, if BagIt (or other standards) was considered for adoption but had to be rejected, this would deserve a more substantive case than provided in the revision.

However, overall while addressing both of these issues would improve the scholarly clarity and value of the manuscript, I don't feel these rise to the level of preventing publication if left as is.

Experimental design

N.A.

Validity of the findings

Some of the features of the reported software continue to be described in the Methods and Results sections interpretative ("easy", "clear", "powerful", etc) rather than strictly fact-based. See comment under (1).

The case for de-facto rejecting BagIt as the underlying dataset structure as currently described is weak and, IMHO, unconvincing. See comment under (1).

Reviewer 2 ·

Basic reporting

This article is much improved. My concerns around references in the introduction, organization of the paper and discussion of the tools use have been addressed. In particular, I appreciate the Use Cases and the new figures. I still would like to have seen more discussion of how dtool is being used at JIC. I think there's a lot of valuable information there on social and/or technical barriers to use, not only of the tool, but of the concept of data management. Given the close association of the tool developers with its users, this is a good opportunity. As the authors continue to develop and deploy dtool, I hope that they will do some 'user testing' and incorporate it into their development and documentation. I look forward to their next paper that reports on further uptake and barriers encountered. :) The authors mention the development of a GUI to help with uptake. I'm not convinced that's the best use of resources, so some analysis before embarking on significant GUI development would be valuable.

Experimental design

Methodology section now address methods of development and design choices.

Validity of the findings

The use cases and Uptake at JIC sections now address how the tool is used. As mentioned above, I would have liked to see more about how it is being used at JIC, however deployment was not set up to collect that data. In the 'Uptake at JIC' section, perhaps the authors could add a sentence to that effect and that they plan to continue to evaluate as they deploy in more labs. Something along the lines of "We did not initially plan deployment to systematically collect information on usage, strengths and barriers to use. As we continue to use and deploy dtool at JIC, we will more systematically collect information and work with users to continue to improve dtool."

Additional comments

Thank you for so thoughtfully addressing our comments. The paper is much improved.

---

## Round 0.3 · accepted · Accept

Thank you for bearing with multiple rounds of revision.

#